# BDNF Outperforms TrkB Agonist 7,8,3′-THF in Preserving the Auditory Nerve in Deafened Guinea Pigs

**DOI:** 10.3390/brainsci10110787

**Published:** 2020-10-28

**Authors:** Henk A. Vink, Willem C. van Dorp, Hans G. X. M. Thomeer, Huib Versnel, Dyan Ramekers

**Affiliations:** 1Department of Otorhinolaryngology and Head & Neck Surgery, University Medical Center Utrecht, Utrecht University, Room G.02.531, P.O. Box 85500, 3508 GA Utrecht, The Netherlands; d.ramekers@umcutrecht.nl (D.R.); h.g.x.m.thomeer@umcutrecht.nl (H.G.X.M.T.); h.a.vink@umcutrecht.nl (H.A.V.); wimkees@gmail.com (W.C.v.D.); 2UMC Utrecht Brain Center, Utrecht University, Universiteitsweg 100, 3584 CG Utrecht, The Netherlands

**Keywords:** cochlea, hearing loss, neurodegeneration, spiral ganglion cell, eCAP, IPG effect, neuroprotection, neurotrophic factor, neurostimulation, small molecule

## Abstract

In deaf subjects using a cochlear implant (CI) for hearing restoration, the auditory nerve is subject to degeneration, which may negatively impact CI effectiveness. This nerve degeneration can be reduced by neurotrophic treatment. Here, we compare the preservative effects of the naturally occurring tyrosine receptor kinase B (TrkB) agonist brain-derived neurotrophic factor (BDNF) and the small-molecule TrkB agonist 7,8,3′-trihydroxyflavone (THF) on the auditory nerve in deafened guinea pigs. THF may be more effective than BDNF throughout the cochlea because of better pharmacokinetic properties. The neurotrophic compounds were delivered by placement of a gelatin sponge on the perforated round window membrane. To complement the histology of spiral ganglion cells (SGCs), electrically evoked compound action potential (eCAP) recordings were performed four weeks after treatment initiation. We analyzed the eCAP inter-phase gap (IPG) effect and measures derived from pulse-train evoked eCAPs, both indicative of SGC healthiness. BDNF but not THF yielded a significantly higher survival of SGCs in the basal cochlear turn than untreated controls. Regarding IPG effect and pulse-train responses, the BDNF-treated animals exhibited more normal responses than both untreated and THF-treated animals. We have thus confirmed the protective effect of BDNF, but we have not confirmed previously reported protective effects of THF with our clinically applicable delivery method.

## 1. Introduction

Damage to or loss of cochlear hair cells in the organ of Corti leads to sensorineural hearing loss (SNHL). In case of severe to profound SNHL, hearing can be partially restored by a cochlear implant (CI), which provides direct electrical stimulation of the spiral ganglion cells (SGCs) that make up the auditory nerve, bypassing the impaired and/or lost hair cells. As the organ of Corti also maintains SGCs by means of neurotrophic support [1,2,3], damage to this structure, including loss of hair cells, leads to degeneration of SGCs [4,5,6,7,8,9]. There is evidence that in CI users, the health of the auditory nerve is compromised, which may negatively impact CI effectiveness [10,11]. Therefore, great efforts in animal studies have been made to prevent nerve degeneration with the aim of improving the hearing of CI users.

In order to prevent SGC degeneration, many studies have investigated the effect of exogenous administration of neurotrophins, in particular the naturally occurring neurotrophin brain-derived neurotrophic factor (BDNF), which significantly enhances SGC survival in deafened guinea pigs (e.g., [1,12,13,14,15,16,17,18,19,20]), cats [21,22] and rats [23]. These animal studies pave the way for possible human applications to benefit CI users [24,25], or as a potential treatment to counteract synaptopathy [26,27]. Importantly, treatment of exogenous BDNF does not only have a positive effect on structural survival of the auditory nerve; its responsiveness to electric current pulses, as provided by a CI, is also significantly improved [14,19,21,28]. In particular, varying the within-subject inter-phase gap (IPG) of these biphasic current pulses leads to changes in the electrically evoked compound action potential (eCAP), which are notably different between deafened animals and normal-hearing animals, indicating a measure of cochlear health [19]. This IPG effect on the eCAP in BDNF-treated deafened guinea pigs has demonstrated near-normal responsiveness of the auditory nerve [19]. The relevance of this IPG measure has recently been shown in several human studies, demonstrating that, unlike absolute eCAP measures, the IPG effect is independent of electrode location [29], and that this within-subject effect may therefore be a potent diagnostic tool (e.g., [30,31]). This indicates that the IPG effect could be as informative of cochlear health in human CI users as it is in guinea pigs, further underlining the clinical relevance of the present study.

The invasive nature of drug delivery to the inner ear by means of a mini-osmotic pump, as often applied in the above-mentioned animal studies, has limited clinical applicability, because of the need for eventual surgical removal, and risk of infection or loss of residual hearing. As a more clinically applicable alternative, local application of BDNF in an absorbable gelatin sponge or hydrogel onto the round window membrane (RWM) has been shown to promote SGC survival as well, albeit not as effectively as with direct infusion into the cochlea [32,33,34,35]. It is thought that this suboptimal effect, which is limited to the basal cochlear turn near the RWM, is partly due to the relatively poor pharmacokinetic properties of BDNF [36,37], as movement throughout the cochlea mainly depends on passive diffusion and large molecules diffuse slowly [38,39].

In cell-based and virtual screenings, small molecules mimicking the effect of BDNF on its target receptor—Tyrosine receptor kinase B (TrkB)—have been discovered [40,41,42,43,44], focusing on the pharmacokinetic and pharmacodynamic profile in comparison to that of BDNF. Yu et al. [45,46] reported similar SGC survival after treatment with the small-molecule TrkB agonists 7,8-dihydroxyflavone (DHF) and 7,8,3′-trihydroxyflavone (THF), to those treated with BDNF in cochlear organotypic cultures. In vivo experiments in mice performed by the same group even showed SGC survival similar to normal-hearing controls [46], further underlining the possible potential of this small-molecule TrkB agonist. Additionally, THF was shown to induce neurite outgrowth in an in vitro study [47], albeit with smaller effects than BDNF.

Since the in vivo results by Yu et al. [46] were very promising in mice, the present study examined the preservative effect of THF on SGCs, compared to that of BDNF. These neurotrophic compounds were applied by means of an absorbable gelatin sponge on the RWM, the same delivery method as in Yu et al. [46], in a guinea pig model of ototoxic deafness. SGC preservation was determined by (1) numerical SGC survival and (2) functional assessment of SGC by means of eCAP recordings. We hypothesized that, compared to BDNF, THF has better pharmacokinetic and pharmacodynamic properties, and will consequently lead to better preservation and functionality of SGCs throughout the cochlea compared to BDNF. Instead, however, our results show that BDNF outperforms THF in both structural and functional preservation of the auditory nerve.

## 2. Materials and Methods

### 2.1. Animals and Experimental Design

Forty-five female guinea pigs (Dunkin Hartley; Hsd Poc:DH; 250–350 g) were obtained from Envigo (Horst, the Netherlands) and kept under standard laboratory conditions (food and water ad libitum; lights on between 7:00 a.m. and 7:00 p.m.; temperature 21 °C; humidity 60%). All animals had normal hearing prior to any experimental procedure, as assessed with click-evoked auditory brainstem responses (ABRs). A schematic overview of the experimental procedure is shown in Figure 1. The animals were divided into one group of normal hearing controls (NH; *n* = 9), and three experimental groups that were ototoxically deafened via systemic co-administration of kanamycin and furosemide. Two weeks after deafening, the animals underwent surgery to receive a gelatin sponge soaked in phosphate-buffered saline (PBS) containing either BDNF (*n* = 12) or THF (*n* = 11) on the perforated round window membrane of their right ear; a negative control group received PBS only (6WD; *n* = 13). Four weeks after deafening, the animals received an intracochlear electrode array in their treated ear with which eCAP recordings were performed before the animals were sacrificed. Both the treated and contralateral untreated cochlea were processed for histology immediately following termination. All surgical and experimental procedures were approved by the Dutch Central Authority for Scientific Procedures on Animals (CCD: 11500201550 and 1150020174315).

### 2.2. Surgical Procedures

#### 2.2.1. Deafening Procedure

Prior to the deafening procedure, the animals were anesthetized by intramuscular injection of dexmedetomidine (Dexdomitor; Vetoquinol, Breda, the Netherlands; 0.25 mg/kg) and ketamine (Narketan; Vetoquinol; 40 mg/kg) followed by pre-operative analgesia by subcutaneous injection of carprofen (Carporal; Dechra/AST Farma, Oudewater, the Netherlands; 4 mg/kg), and click-evoked ABRs were recorded to verify normal hearing (see [48] for details). Thresholds <40 dB peak equivalent SPL were considered to indicate normal hearing. Deafening was done by subcutaneous injection of kanamycin (Sigma-Aldrich, St. Louis, MO, USA; 400 mg/kg) followed by infusion of furosemide (Centrafarm, Etten-Leur, the Netherlands; 100 mg/kg) into the external jugular vein, which has been shown to reliably eliminate the majority of both inner and outer hair cells [9,49].

#### 2.2.2. Gelatin Sponge-Mediated Treatment

Two weeks after the deafening procedure, the animals were again anesthetized with ketamine and dexmedetomidine, and were given carprofen as pre-operative analgesia, followed by recordings of click-evoked ABRs to confirm the animals were sufficiently deafened. Animals with a threshold shift of >50 dB were considered successfully deafened and were included in the study. Subsequently, the right cochlear bulla was exposed via a retro-auricular approach. A small hole was made in the bulla to expose the round window membrane (RWM) of the cochlea. A micropick was used to perforate the RWM before a small absorbable hemostatic gelatin sponge (Spongostan™ Dental; Ethicon, Somerville, NJ, USA) cylinder (~1 mm^3^), soaked in 3 µL of the treatment solution, was placed into the round window niche (see [35], their Figure 1). The treatment solution consisted of PBS with 15% dimethyl sulfoxide (DMSO) containing either BDNF (PeproTech, London, UK; 6.67 µg/µL (240 µM) in PBS) or 7,8,3′-trihydroxyflavone (Santa Cruz Biotechnology, Inc., Heidelberg, Germany; 0.0667 μg/μL (247 µM)). The addition of 15% DMSO was required for THF to be dissolved, and this addition was maintained for all deafened groups for the purpose of comparability. After the gelatin sponge placement, the bulla was sealed with dental cement (GC Fuji PLUS; GC Corporation, Tokyo, Japan).

#### 2.2.3. Acute Implantation

Four weeks after the gelatin sponge placement, anesthesia was induced by injection of Hypnorm^®^ (Vetapharma; 0.5 mL/kg i.m.) and subsequent administration of a gas mixture of 2% isoflurane evaporated in O_2_ and N_2_O (1:2) via a mouth cap. The skull was exposed and two transcranial screws were placed 1 cm bilateral to bregma to serve as reference electrodes for eCAP recordings. The animals were subsequently tracheostomized and artificially ventilated (Amsterdam infant ventilator mk3, Hoekloos, Schiedam, the Netherlands) with a gas mixture of O_2_ and N_2_O (1:2) and 1–1.5 % isoflurane (45–50 cycles/min respiration rate, 1.8–2.3 kPa) throughout the remainder of the experiment. The bulla was reopened to expose the cochlea. A 0.5-mm cochleostomy was then drilled in the basal turn, within 1 mm from the round window, through which a custom-made four-contact electrode array was inserted into the scala tympani. This array was connected to a MED-EL PULSAR cochlear implant (MED-EL GmbH, Innsbruck, Austria).

### 2.3. Electrophysiology

#### 2.3.1. ABR Recordings

Click-evoked ABRs were recorded both prior to ototoxic deafening and prior to treatment initiation using three subcutaneous needle electrode: (1) the ground electrode was placed in the hind limb, (2) the active electrode behind the right ear and (3) the reference electrode was placed on the skull, rostral to the brain. Broadband acoustic clicks (20 μs monophasic rectangular pulses; inter-stimulus interval 99 ms) were synthesized and attenuated using a TDT3 system (Multi-I/O processor RZ6; Tucker-Davis Technologies, Alachua, FL, USA), and presented in free field, at 10 cm distance from the right ear, using a Blaupunkt speaker (PCxb352; 4 Ω; 30 W, Blaupunkt (International) GmbH & Co. KG, Hildesheim, Germany). For amplification of the ABRs, a Princeton Applied Research (Oak Ridge, TN, USA) 5113 pre-amplifier (amplification ×5000; band pass filter 0.1–10 kHz) was used. The response was subsequently digitized by the TDT3 system (100 kHz sampling rate, 24-bit sigma-delta converter), and stored on a PC for offline analysis. Hearing thresholds were determined by starting at 110 dB peak equivalent SPL and subsequently decreasing the sound level by 10 dB until no further response was observed. The threshold was then defined as the interpolated sound level at which the ABR N_1_-P_2_ peak was 0.3 µV.

#### 2.3.2. eCAP Recordings

Two protocols of eCAP recordings were performed as described by Ramekers et al. ([48], single-pulse protocol; [19], pulse-train protocol). The implant was controlled by a PC via a Research Interface Box 2 (Department of Ion Physics and Applied Physics, University of Innsbruck, Innsbruck, Austria) and a National Instruments data acquisition card (PCI-6533, National Instruments). Stimulation/recording paradigms were executed using MATLAB (version 7.11.0; Mathworks, Natick, MA, USA). Typically, the most apical of the four electrodes on the intracochlear array was used for stimulation (~4 mm inside the cochleostomy) and the most basal one was used for recording. For both protocols, biphasic current pulses of 30 μs/phase were presented with alternating polarity to reduce stimulation artifact, and the responses to 50 pairs of these stimuli were averaged.

For the single-pulse recordings, the inter-phase gap (IPG) of the stimuli was either 2.1 or 30 μs. Varying this IPG leads to differences in the eCAP, which are found to be indicative of neural health [48], as depicted in Figure 2A. For both IPGs the pulse was presented at 20 current levels, typically ranging from 40 µA to 800 µA, with a maximum current level ranging from 800 µA to 1100 µA (800: *n* = 24; 850: *n* = 8; 900: *n* = 4; 950: *n* = 2; 1100: *n* = 1).

Pulse trains consisted of series of ten identical biphasic current pulses with 30-µs IPG at the highest current level used for the single-pulse recordings. In order to be able to record the responses to each pulse in the 10-pulse pulse train, eCAPs were recorded in ten steps, in which the number of stimulus pulses increased from 1 to 10 (see Figure 3A). The inter-pulse intervals (IPIs) were thus chosen so that the longest (16 ms) does not cause any refractoriness or adaptation (i.e., every consecutive pulse evokes a similarly large response), while the shortest (0.4 ms) leads to a transient disappearance of the eCAP caused by mechanisms such as refractoriness, neural fatigue or adaptation (Figure 3B).

### 2.4. Histological Processing

Both treated and the untreated contralateral cochleas were extracted after termination. The cochleas were fixated by an intra-labyrinthine infusion with a fixative of 3% glutaraldehyde, 2% formaldehyde, 1% acrolein and 2.5% dimethyl sulfoxide (DMSO) in a 0.08 M sodium cacodylate buffer, as described by De Groot et al. [50]. The cochleas were decalcified, post-fixated and embedded in Spurr’s low-viscosity resin. Staining was performed using 1% methylene blue, 1% azur B and 1% borax in distilled water. The cochleas were subsequently divided into two halves along a standardized midmodiolar plane and then re-embedded in fresh resin. From one of these halves, two semithin (1 μm) sections were cut at least at 60 μm intervals, which were used for the quantification of SGCs.

### 2.5. Data Analysis

#### 2.5.1. Electrophysiological Analyses

For the single-pulse analysis, the eCAP amplitude—defined as the voltage difference between the N_1_ and P_2_ peaks (Figure 2A)—was plotted against current level, resulting in an input–output function, as exemplified in Figure 2B. These were subsequently fitted with a Boltzmann sigmoid as in Ramekers et al. [48]
(1)$$V_{eCAP}=A+\;\frac{B}{1+e^{-\frac{I-C}{D}}},$$
in which V_eCAP_ is amplitude in µV, I is stimulation current in µA and A–D are fitting parameters. From the fitting parameters, several measures can be derived (see Figure 2B): the maximum N_1_-P_2_ amplitude (B), the current to reach the half-maximum amplitude (C), slope at C (B/4D), threshold (C−2D) and the dynamic range (4D). The N_1_ peak latency, averaged over the three highest current levels, was analyzed in addition to these input–output characteristics.

The analysis of eCAPs in response to pulse trains was largely done as described by Ramekers et al. [51]. As the eCAP N_1_-P_2_ amplitude decreased with decreasing IPI (Figure 3B), an alternating pattern emerged (Figure 3A,C), the amplitude of which has been previously shown to be inversely correlated to SGC survival [19,51]. In order to only include the stable alternating pattern at the end of the pulse train, the responses to the first four pulses were excluded (Figure 3C). The amplitude of this modulation between odd- and even-numbered pulses was expressed as a relative measure by dividing it by the maximum eCAP amplitude (as evoked by the first of the ten pulses) for each individual animal. All analyses were performed using custom-made software in MATLAB.

#### 2.5.2. Histological Analyses

A Leica DC300F digital camera mounted on a Leica DMRA light microscope was used with a 40x oil immersion objective (Leica Microsystems GmbH, Wetzlar, Germany) to obtain micrographs of Rosenthal’s canal of the cochlear regions B1, B2, M1, M2, A1, A2 and A3 (Figure 4A) from the two sections per cochlea. The A3 region of both the right and left cochlea could only be acquired for 17 out of 36 animals; for all other regions this was least 33 out of 36. Subsequently, the packing density and perikaryal area of both type I and type II SGCs was established for cochlear regions B1 t/m A2 using ImageJ (version 1.52a; National Institutes of Health, Bethesda, MD, USA) by outlining the bony boundaries of Rosenthal’s canal and quantifying the SGCs within it (Figure 4B,C). The packing density was then averaged across the two sections. As SGC size may influence the likelihood of detecting it, the obtained packing density was corrected for the mean perikaryal area, as previously described [52,53]. Analyses were performed for all deafened animals and six NH animals.

#### 2.5.3. Statistics

Linear mixed model (LMM) analyses were applied under the assumption of compound symmetry, in the treated right cochlea with cochlear location as a covariate and treatment (BDNF vs. THF; BDNF vs. 6WD or THF vs. 6WD) as a factor. In this analysis, location was expressed in relative distance from location B1 as described by Van Loon et al. ([53], their Figure 2). Differences between the treated ears and contralateral ears were assessed by performing a log_2_ transformation on the right (AD)/left (AS) ratio and testing this against 0 with one-sample *t*-tests for each group for every cochlear location.

Group differences in eCAP characteristics were assessed with independent samples t-test or one-way ANOVA, followed by post hoc Tukey’s HSD. Pearson’s correlation coefficients were calculated to elucidate the relationship between the single pulse eCAP characteristics and pulse train response parameters with SGC survival.

The LMM analyses, ANOVA and one-sample *t*-tests were performed in SPSS Statistics 25.0.0.2 for windows (IBM Corp. Armonk, NY, USA). Pearson’s correlation coefficients were calculated using MATLAB (version 9.1.0, Mathworks, Natick, MA, USA).

## 3. Results

### 3.1. Animal Inclusion

All animals were confirmed to be normal-hearing prior to the experiment. The animals from the three deafened groups (BDNF-treated, THF-treated, and untreated 6WD) were successfully deafened, with an average ABR threshold shift of 76 dB (ranging from 66 to 83 dB). One BDNF-treated animal suffered fatal respiratory complications during preparation for the acute eCAP recordings; hence, for this animal, only histological data were available. For all other animals, all eCAP recordings were performed successfully. However, in one 6WD animal and one THF-treated animal it was not possible to reliably fit a Boltzmann curve to their amplitude growth functions, and therefore only their eCAP latency was included in the single-pulse analyses.

### 3.2. Spiral Ganglion Cell Survival

Typical micrograph examples of SGCs in the cochlear B1 region of an animal from each of the four experimental groups are shown in Figure 5. In a normal-hearing animal (Figure 5A), Rosenthal’s canal is densely packed with SGCs, whereas six weeks after deafening (Figure 5B), a strong decrease in cells can be observed and the remaining cells appear to be smaller than in the normal-hearing condition. Following BDNF treatment (Figure 5C), SGC loss can be observed but cells appear more densely packed than without treatment, and the SGCs appear more like healthy cells. After THF treatment (Figure 5D), degeneration is apparent, although there appear to be more SGCs present than without treatment, but with similar morphology.

In Figure 6A, normalized mean SGC packing densities are shown per group as a function of cochlear location in relative distance from the round window (absolute values for normal-hearing animals in Figure 6A; inset). When comparing the right cochleas between the experimental groups, the SGC packing density of the BDNF treatment group showed higher SGC survival in cochlear locations B1 and B2 than in the untreated 6WD group, but became virtually indistinguishable from the 6WD group from location M1 onwards. Statistically, BDNF-treated cochleas had a significantly higher SGC packing density than the untreated 6WD cochleas (linear mixed model; *F*_(1,67.2)_ = 6.3; *p* = 0.015) with a significant interaction effect between group and cochlear location confirming the diminishing difference between the two towards the cochlear apex (linear mixed model; *F*_(1,77.7)_ = 13.8; *p* < 0.001). In contrast, although THF-treated animals showed more remaining SGCs in the B1 region than the untreated 6WD animals, the effect of THF treatment was not statistically significant (linear mixed model; *F*_(1,41.9)_ = 0.26; *p* = 0.61), neither was there an interaction effect between treatment and location. The difference between the BDNF and THF treatment groups was not statistically significant (linear mixed model; *F*_(1,41.3)_ = 2.1, *p* = 0.15). Surprisingly, the contralateral ears in the BDNF group showed significantly less SGC survival than in the THF group (linear mixed model; *F*_(1,44.5)_ = 4.2, *p* = 0.046) but not less than the 6WD control group (linear; *F*_(1,36.7)_ = 1.7, *p* = 0.20).

As BDNF treatment led to more SGC survival than the 6WD control animals the differences between treated right and untreated left ears (visualized in Figure 6B as log_2_ transformed right/left SGC survival) were further investigated. SGC survival in the right treated ear of the BDNF animals was more than twice as high as in their contralateral ear in cochlear location B1 (one-sample t-test; B1: *t*_(11)_ = 6.8; *p* < 0.001). The difference decreased towards the apex with, significantly higher SGC survival in cochlear locations B2 and M1 (one-sample *t*-test; B2: *t*_(11)_ = 2.5; *p* = 0.031; M1: *t*_(11)_ = 2.3; *p* = 0.040) and no difference was observed in the more apical cochlear locations.

To rule out the effect of cochlear manipulation (gelatin-sponge placement, RWM perforation, CI insertion) on SGC survival, right and left ears of the untreated 6WD animals were compared (Figure 6B). Substantial SGC loss had occurred (~50% re NH), which did not statistically differ between the two ears for any cochlear location (*p* > 0.3).

### 3.3. Responsiveness of Spiral Ganglion Cells to Electrical Stimuli (eCAPs)

Treatment with BDNF, but not with THF, yielded enhanced survival of SGCs. To examine whether BDNF treatment additionally resulted in improved neural responsiveness to electrical stimulation and to examine whether THF treatment may affect responsiveness in spite of the limited effect on SGC survival, we investigated several eCAP measures of the experimental groups.

#### 3.3.1. Absolute eCAP Characteristics and the IPG Effect

As exemplified in Figure 2, increasing the IPG of the electric current pulse from 2.1 to 30 µs leads to differences in eCAP characteristics. In NH animals, this results in an increase in N_1_-P_2_ peak amplitude, a steeper slope, a lower threshold, a wider dynamic range, and a lower level_50%_, while latency remains unaffected by varying the IPG. Figure 7 shows the group averages of the eCAP input–output characteristics. Note that, in this figure, the results from the NH animals and those from the two-weeks-deaf animals (2WD; previously reported by [48]) are shown for reference purposes. Only the 6-weeks-deaf experimental groups were used for subsequent statistical analyses, as the eCAP measures of untreated 6WD animals generally significantly differ from those of NH and 2WD animals [48].

When assessing the absolute eCAP characteristics only for slope (Figure 7D,E) and dynamic range (Figure 7J,K) statistical differences were observed between the groups (Table 1). At an IPG of 30 µs, the BDNF-treated animals had a steeper slope than the untreated 6WD animals (Tukey’s HSD; *p* = 0.025). Additionally, the BDNF treatment group had a normal-like dynamic range, at an IPG of 30 µs, which was significantly narrower than that for both the untreated 6WD and THF treatment groups (Tukey’s HSD; *p*_BDNF-6WD_ = 0.004; *p*_BDNF-THF_ = 0.01). THF did not appear to influence SGC responsiveness as the treatment group did not statistically differ from the untreated 6WD group for any of the absolute eCAP characteristics (Figure 7, first two columns).

When examining the IPG effect, differences in ∆amplitude (Figure 7C), ∆threshold (Figure 7I), ∆dynamic range (Figure 7L), and ∆level_50%_ (Figure 7O) were observed between the experimental groups (Figure 7 third column; Table 1). Post hoc analyses revealed that the ∆amplitude and ∆dynamic range of the BDNF treatment group were significantly lower and more normal than that of the untreated 6WD animals (Tukey’s HSD; amplitude: *p* = 0.006; dynamic range: *p* < 0.001). Additionally, although both THF and BDNF treatment resulted in a higher ∆threshold than 6WD, this was only significant for BDNF (Tukey’s HSD; *p*_BDNF-6WD_ = 0.020; *p*_THF-6WD_ = 0.063). Finally, while for ∆level_50%_ neither the BDNF- nor the THF-treated animals differed significantly from the untreated animals, ∆level_50%_ was smaller after treatment with BDNF than with THF (Tukey’s HSD; *p* = 0.036). No differences were observed between the experimental groups for ∆slope (*p* = 0.38; Figure 7F) or latency (*p* > 0.6; Figure 7P–R).

#### 3.3.2. Single Pulse eCAP Characteristics and SGC Survival

As the strongest treatment effects on SGC survival were observed in the B1 and B2 regions of the cochlea, we averaged the SGC packing densities of B1 and B2 into one value representing the basal SGC packing density. Subsequently, in order to explore the relationship between the eCAP characteristics and remaining SGCs, both the absolute eCAP characteristics and the IPG effect for all six eCAP characteristics are presented as a function of basal SGC survival for the individual animals of the BDNF and THF treatment groups (Figure 8).

The NH and untreated 6WD animals (from the present study) and the 2WD animals (from [48]) are represented by the black regression lines. Significant correlations were observed between most of the absolute eCAP characteristics and the IPG effect, and SGC cell packing density, shown as solid lines in Figure 8. The non-significant correlations (Figure 8G,H,J,K,M,O) are shown as dashed lines. *R*^2^ and *p* values of these correlations are presented in Table 2.

For the absolute amplitude (Figure 8A,B), with both IPGs, the animals from both the BDNF and THF treatment groups did not appear to follow the regression line, indicating that the basal SGC survival in these animals did not greatly affect the maximum eCAP amplitude. Similarly, threshold (Figure 8G,H), dynamic range (Figure 8J,K), and level_50%_ (at an IPG of 2.1 µs; Figure 8M,N), appeared to be largely unaffected by the number of surviving SGCs in all treated animals. For slope (Figure 8D,E), the treated animals are in close proximity to the regression line. The individual variation in latency in both the BDNF and THF groups indicates no clear relationship between the SGC preservation by the treatment and this eCAP characteristic.

For the ∆amplitude, ∆slope, ∆threshold and ∆dynamic range (Figure 8C,F,I,L) the BDNF- and THF-treated animals are close to the regression line. For ∆amplitude (Figure 8C), ∆dynamic range (Figure 8L) and ∆level_50%_ (Figure 8O), most of the BDNF-treated animals and some of the THF-treated animals are situated beneath the line, indicating a more normal functionality, regardless of the number of SGCs.

In summary, treatment with BDNF not only led to numerical but also to functional preservation of SGCs. However, functional preservation of SGCs following THF treatment, as evaluated by the six eCAP characteristics, was not observed.

#### 3.3.3. Pulse Trains

Two eCAP characteristics derived from pulse train stimulation are shown in Figure 9. The normalized amplitude modulation (from the alternating N_1_-P_2_ amplitude, as exemplified in Figure 3A,C) became stronger with a decrease in inter-pulse interval (IPI; Figure 9A). This was more pronounced in the deafened animals, showing the highest modulation at IPIs of 0.4 and 0.6 ms. For an IPI of 0.6 ms, as previously shown by Ramekers et al. [51], the modulation was significantly higher for the untreated 6WD animals than NH animals (independent samples *t*-test; *t*_(16)_ = 5.0, *p* < 0.001). At this IPI, the modulation for both the BDNF and the THF treatment groups was lower than that of the untreated 6WD group, but not significantly so (one-way ANOVA; *F*_(2,28)_ = 3.2, *p* = 0.054). The N_1_ latency, shown in Figure 9B as the mean latency of the eCAPs evoked by the last six pulses in the pulse train, increased with decreasing IPI for all groups. The ∆N_1_ latency (difference in N_1_ latency between an IPI of 0.6 and 16 ms; see dashed boxes in Figure 9B) was smaller in the untreated 6WD group than in the NH group (independent samples *t*-test; *t*_(16)_ = 4.1, *p* < 0.001). Among the three deafened groups, differences in ∆N_1_ latency were observed (one-way ANOVA; *F*_(2,28)_ = 4.4, *p* = 0.022), with the BDNF treatment group having a significantly higher ∆N_1_ latency than the untreated 6WD group, without differing from the THF treatment group (Tukey’s HSD; *p*_BDNF-6WD_ = 0.028 and *p*_BDNF-THF_ = 0.95) and the THF treatment group having a higher, but not statistically different, ∆N_1_ latency than the untreated 6WD group (Tukey’s HSD; *p* = 0.053).

#### 3.3.4. Pulse Trains and SGC Survival

To assess the relation between pulse train responses and SGC survival, both the amplitude modulation for each individual animal at an IPI of 0.6 ms as well as the ∆N_1_ latency are shown as a function of basal SGC survival (Figure 9C,D, respectively).

For the untreated animals (NH and 6WD), amplitude modulation was significantly negatively correlated with basal SGC survival (Figure 9C). The amplitude modulation in the NH animals was clearly lower than that in the untreated 6WD animals, whereas both the BDNF- and THF-treated animals appeared to perform as well as the most normal-like untreated 6WD animals. The BDNF treatment animals were generally in close proximity to the regression line, exhibiting a relation between eCAP modulation and basal SGC survival. However, whereas, the averaged values of the THF-treated group are close to the regression line, the individuals of this group do not follow this line, clearly indicating no relationship between eCAP modulation and basal SGC survival.

The ∆N_1_ latency for the untreated animals (NH and 6WD) strongly correlated with basal SGC survival (Figure 9D). Furthermore, in contrast to the amplitude modulation (Figure 9C), the ∆N_1_ latency regression more clearly represented the individual animals of both the untreated and the treatment groups. This suggests that ∆N_1_ latency can be used as an accurate indicator for SGC survival, irrespective of treatment.

In summary, in line with our observations of the IPG effect (Section 3.3.1 and Section 3.3.2), the BDNF-treated animals exhibited more normal pulse-train-evoked responses than both untreated and THF-treated animals.

## 4. Discussion

In the present study, we compared the effects of BDNF and 7,8,3′-THF on SGC survival and neural responsiveness in a guinea pig model of sensorineural hearing loss, when administered to the cochlea in a clinically feasible fashion by means of a gelatin sponge on the round window membrane, a similar approach as used by Havenith et al. [34,35] and Yu et al. [46]. Based on these previous reports, we expected both compounds to significantly preserve SGCs and positively affect neural function up to four weeks after treatment. Treatment with BDNF indeed led to significant SGC survival and improved neural responsiveness, whereas THF did not.

### 4.1. BDNF, but not THF, Promotes SGC Survival

Yu et al. [46] compared the neurotrophic potential of THF with that of BDNF, reporting significant SGC survival in vitro which was similar after treatment with either BDNF or THF. Moreover, in vivo, they found significant enhancement of SGC survival in the basal cochlear turn in cCx26 null mice 30 days after gelatin-sponge-mediated delivery of THF, which was ~3.5 times higher than their untreated controls. This sparked our interest and led to our choice to investigate THF as a small-molecule substitute for BDNF.

The presently observed SGC survival following BDNF treatment is in line with our previous studies using gelatin sponge application of BDNF [34,35], in which survival was apparent but limited to the lower basal turn. In the present study, enhanced SGC preservation was observed up to the lower middle turn, most probably because the amount of BDNF used (20 µg) was three times higher compared to the aforementioned studies (6 µg). The same BDNF dosage (20 µg) has been used in our lab in previous studies, using mini-osmotic pumps as a delivery method [19,28,54], in which the preservative effect extended up to the apical turns. This difference indicates that the delivery method strongly influences treatment effectiveness, with gelatin-sponge-mediated delivery not being as effective as direct cochlear infusion. The present results also suggest that an increase in BDNF dose may further increase effectiveness of the present method.

In sharp contrast, however, the reported in vivo protective effect of THF was not confirmed in our deafened guinea pig model, as THF did not yield any positive effect on SGC survival. Both compared to the sham-treated 6WD group (Figure 6A), as well as individually compared to their left untreated ears (Figure 6B), we did not observe enhanced survival of SGCs in the THF-treated cochleas.

### 4.2. Functional Preservation after BDNF Treatment

We have previously shown that all eCAP characteristics assessed in the present study are affected by IPG variation [19,48]. Following Prado-Guitierrez et al. [55], we subsequently confirmed dependency of this IPG effect on SGC survival. Accordingly, for the untreated normal-hearing and deafened animals in the present study, this effect was visible for all six eCAP characteristics (Figure 7, right column). BDNF treatment resulted in more normal responsiveness than either THF treatment or the 6WD group, for ∆maximum amplitude, ∆threshold, ∆dynamic range and ∆level_50%_, as illustrated by their similarity to the NH values (Figure 7C,I,L,O). The N_1_-P_2_ maximum amplitude increased with the increase in the IPG of the biphasic current pulse, while the stimulation threshold decreased. This effect is larger in deafened than in NH animals, possibly because the degenerating SGCs have a higher excitation threshold and require more time to initiate and propagate an action potential [48], thus benefiting more from an increase in IPG than healthy SGCs, which respond almost as well to current pulses with a short IPG as to pulses with a long IPG. The increase in dynamic range in deafened animals, following the increase in IPG, is a likely consequence of an SGC population with an increased inter-neuron variation in excitation thresholds, as the cells with a high threshold are only recruited with a longer IPG [48]. The similarity to a normal-hearing response for these eCAP measures after treatment indicates that the BDNF-treated SGCs can still function efficiently with a short IPG, i.e., better than the untreated 6WD animals. Level_50%_, the stimulation level to reach a half-maximum amplitude is dependent on both stimulation threshold and the dynamic range. A decrease in threshold and an increase in dynamic range have opposing effects on level_50%_. This could explain why the BDNF-treated animals significantly differed from the THF-treated animals with respect to ∆level_50%_, but not with respect to ∆threshold or ∆dynamic range.

In short, this suggests that the surviving SGCs, following BDNF treatment, showed functionality similar to healthy cells, even though fewer SGCs were present than in the healthy situation. This is further substantiated by the observed correlation between IPG effects and SGC packing density in the literature [19,48] and in the present study (Figure 8). Ramekers et al. [19] showed a strong positive correlation between the IPG effect of ∆maximum amplitude, ∆latency and ∆level_50%_, and SGC survival after BDNF treatment with an osmotic pump, with BDNF-treated animals exhibiting a normal-like ∆level_50%_ with fewer SGCs. With our gelatin-sponge-mediated delivery, we observed similar results for ∆maximum amplitude and ∆level_50%_ when compared to basal SGC survival alone.

A second measure of cochlear neural health is the eCAP amplitude modulation in response to pulse trains, which has been shown to be indicative of SGC survival in both untreated animals [51] and BDNF-treated animals [19], which is confirmed in the present study. Importantly, whereas in the two aforementioned studies we used relatively long (100 ms) pulse trains, we have shown here that similarly potent cochlear neural health measures can be obtained with pulse trains of <10 ms duration (e.g., 10 pulses of IPI of 0.6 ms). This makes this measure more clinically applicable, not only because of the reduction in time it takes to complete the eCAP recording sessions, but also because high-frequency pulse trains may increase loudness perception to uncomfortable levels [56]. While the group means of BDNF- and THF-treated animals do overlap for amplitude modulation (Figure 9A), this is not reflected in the number of SGCs (Figure 9C). Therefore, it is possible that THF does have an effect on SGC function (i.e., responsiveness to high-frequency stimulation), while otherwise not having an effect on cell survival. Except for this specific observation, we have not been able to find evidence of changed electrophysiology—be it positive or negative—after treatment with THF.

### 4.3. The Elusive Effect of THF

Since we expected THF to at least match the effect of BDNF, the lack of any effect from the THF treatment is surprising. More so since recently, in in vitro studies in mice and rats, THF has been found to be effective in promoting neurite outgrowth in dorsal root ganglion explants [57] and cochlear SGC explants [47] in a reportedly TrkB-dependent manner. Frick et al. [47] compared the effects of BDNF with THF on neurite outgrowth. They reported that while THF promoted neurite outgrowth in spiral ganglion explants of P7 NMRI mouse cochleas, this effect was smaller than with BDNF, which induced outgrowth beyond control level from P3 to P7. As we have not investigated neurite outgrowth in our histological analyses (i.e., assessment of peripheral processes), we cannot corroborate these results. These aforementioned results do suggest that THF has an effect on SGCs which, however, is limited compared to BDNF.

THF might not activate TrkB the same way as BDNF does. Studies investigating DHF, of which THF is a derivative, have reported its binding to the Ig2 region [58] and CC2 region [42] of the TrkB extracellular domain, which overlaps with BDNF binding sites. Additionally, phosphorylation of TrkB and AKT (part of the PI3K-AKT pathway, the secondary downstream pathway associated with cell survival and cell growth) was shown to be induced by DHF, albeit depending on neuronal maturation [59]. In contrast, Boltaev et al. [37] measured TrkB and AKT phosphorylation of DHF, and found that DHF did not result in phosphorylation of either. Although the latter could explain the lack of SGC survival following THF treatment observed in the present study, these contradictory findings illustrate that the effect of THF on TrkB is yet to be fully understood.

### 4.4. Methodological Considerations

The mouse cochlear fluid volume is 10 times smaller than that of a guinea pig cochlea [60] with the mouse cochlea having four half-turns against eight half-turns in the guinea pig ([61], their Figure 1). Therefore, the diffusion of THF molecules throughout the cochlea would be notably different between both species, which could explain why Yu found an effect in mice which is not observable in guinea pigs, and—possibly, by extension—in humans (~60 larger cochlear volume than mice; [60]).

Secondly, we encountered an issue with solubility of THF prior to our experiments, as at least 15% DMSO was required to successfully create a 0.0667 µg/µL THF solution. In contrast, Yu et al. [46] managed to create a similar THF solution (0.054 µg/µL) with only 0.1% DMSO. This strong difference is notable at the very least, and may have affected THF performance in addition to the difference in cochlear volumes. Note that with the same DMSO concentration, BDNF had a substantial neuroprotective effect, which was even stronger than previously reported [34,35]. Moreover, as no differences were found in SGC survival between the treated and contralateral ear in the 6WD animals, it is unlikely that 15% DMSO negatively affected the SGCs. However, using a solution with 15% DMSO to treat humans is undesirable. Therefore, even without considering the better outcome obtained with BDNF, treatment with the latter is preferable.

## 5. Conclusions

We set out to investigate the protective effects of BDNF and THF using a clinically applicable method, as opposed to the more invasive means of cochlear infusion for which it has been established that BDNF positively affects SGC survival and neural function. When applied in a gelatin-sponge-mediated manner, BDNF indeed preserved SGCs and their function, whereas THF did neither, despite previous reported evidence to the contrary. Therefore, focusing on BDNF as a means to preserve the auditory nerve is a prudent course for the future. Additionally, the present study underlined the correlation between several eCAP measures (including the IPG effect) and SGC survival; this is particularly useful as these outcome measures could potentially be used to estimate neural health in human CI users.

## Figures and Tables

**Figure 1 brainsci-10-00787-f001:**
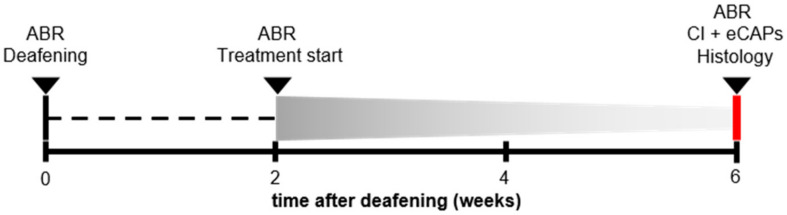
A schematic overview of the experimental procedure. ABR: acoustically evoked auditory brainstem response; CI: cochlear implant; eCAP: electrically evoked compound action potential.

**Figure 2 brainsci-10-00787-f002:**
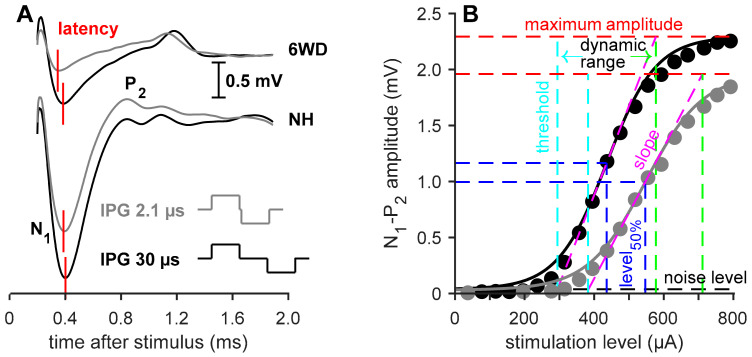
Examples of single pulse eCAP recordings. (**A**) eCAPs from an untreated 6 weeks deaf (6WD) and a normal-hearing (NH) animal with an IPG of 2.1 or 30 µs.**** N_1_ represents the first negative peak and P_2_ represents the second positive peak of the waveform. The red vertical lines indicate the N_1_ latency of the eCAP. (**B**) Input-output functions from N_1_-P_2_ amplitude in the same NH animal of A both an IPG of 2.1 µm (gray) and 30 µs (black). The line represents the Boltzmann curve fit (Equation (1)). The dashed, colored lines represent the various eCAP characteristics derived from the aforementioned Boltzmann equation. IPG: inter-phase gap.

**Figure 3 brainsci-10-00787-f003:**
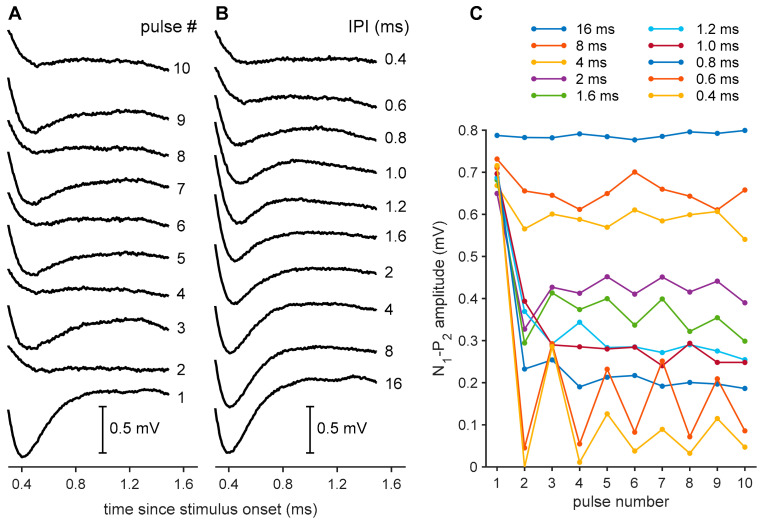
Examples of eCAP recordings to pulse trains from a single brain-derived neurotrophic factor (BDNF)-treated animal. (**A**) eCAPs in response to pulse trains of 1 (lowest trace) to 10 (upper trace) pulses with inter-pulse interval (IPI) of 0.6 ms. An alternating pattern can be observed for odd- and even-numbered pulses. The N_1_-P_2_ peak amplitudes for these eCAPs are shown in (**C**) (0.6 ms trace). (**B**) eCAPs in response to the final (tenth) pulse of pulse trains with various IPIs. The N_1_-P_2_ peak amplitudes for these eCAPs are shown in (**C**) (last datapoints for all traces). (**C**) eCAP N_1_-P_2_ peak amplitudes in response to all 10 pulses in pulse trains with 10 different IPIs.

**Figure 4 brainsci-10-00787-f004:**
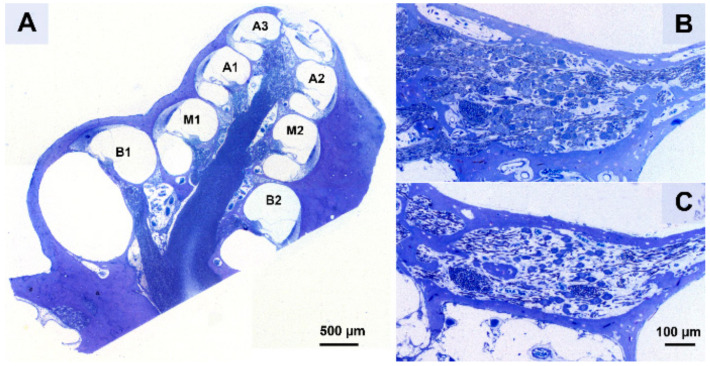
(**A**) A cross-section along a standardized midmodiolar plane. From this cross-section, the spiral ganglion cells (SGCs) within Rosenthal’s canal were quantified for the seven cochlear regions B1–A3. (**B**) An example of Rosenthal’s canal in B1 from a normal-hearing animal (NH) and (**C**) a six weeks deaf untreated animal (6WD).

**Figure 5 brainsci-10-00787-f005:**
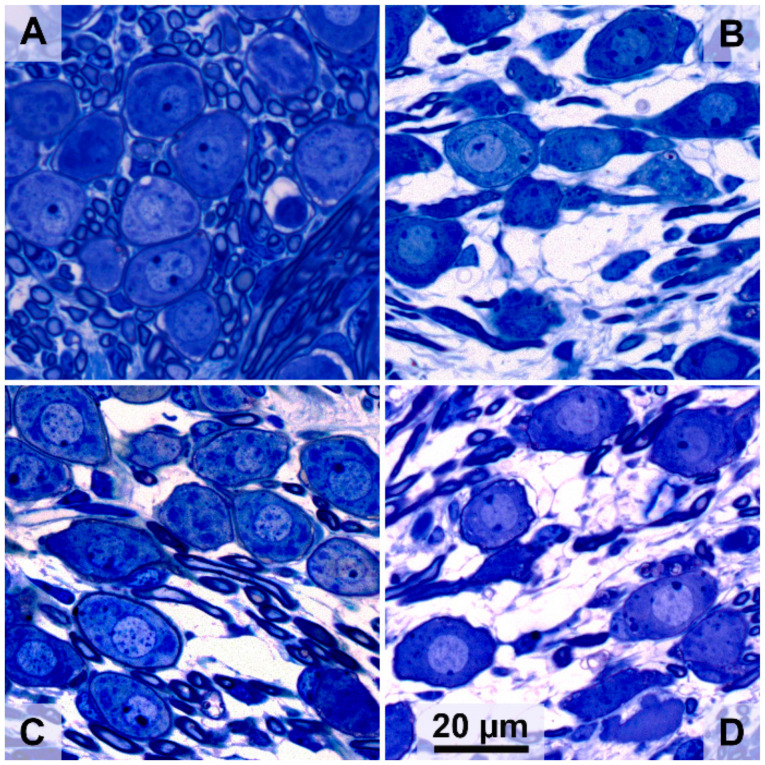
Representative examples of SGCs in Rosenthal’s canal at cochlear location B1 from the treated right ears. (**A**) Normal-hearing animal; (**B**) untreated 6 weeks deaf (6WD) animal; (**C**) BDNF-treated animal; (**D**) 7,8,3′-trihydroxyflavone (THF)-treated animal.

**Figure 6 brainsci-10-00787-f006:**
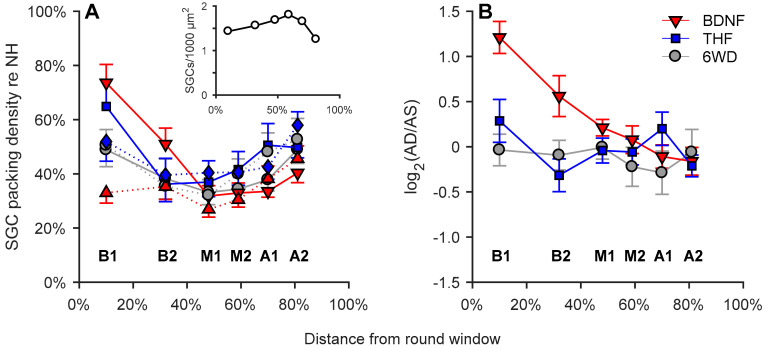
Normalized SGC packing density as a function of cochlear location in relative distance from the round window, in which 0% is the round window and 100% is the helicotrema. (**A**) Group means of SGC packing density plotted separately for right treated ears (AD, solid lines) and left untreated ears (AS, dashed lines). BDNF_AD_ = down-pointing triangle; BDNF_AS_ = up-pointing triangle; THF_AD_ = square; THF_AS_ = diamond. (**B**) Log transformed SGC packing density ratio AD/AS. NH, *n* = 6; 6WD, *n* = 13 BDNF, *n* = 12; THF, *n* = 11. B1 to A2 represent the cochlear locations from base to apex. Error bars represent SEM.

**Figure 7 brainsci-10-00787-f007:**
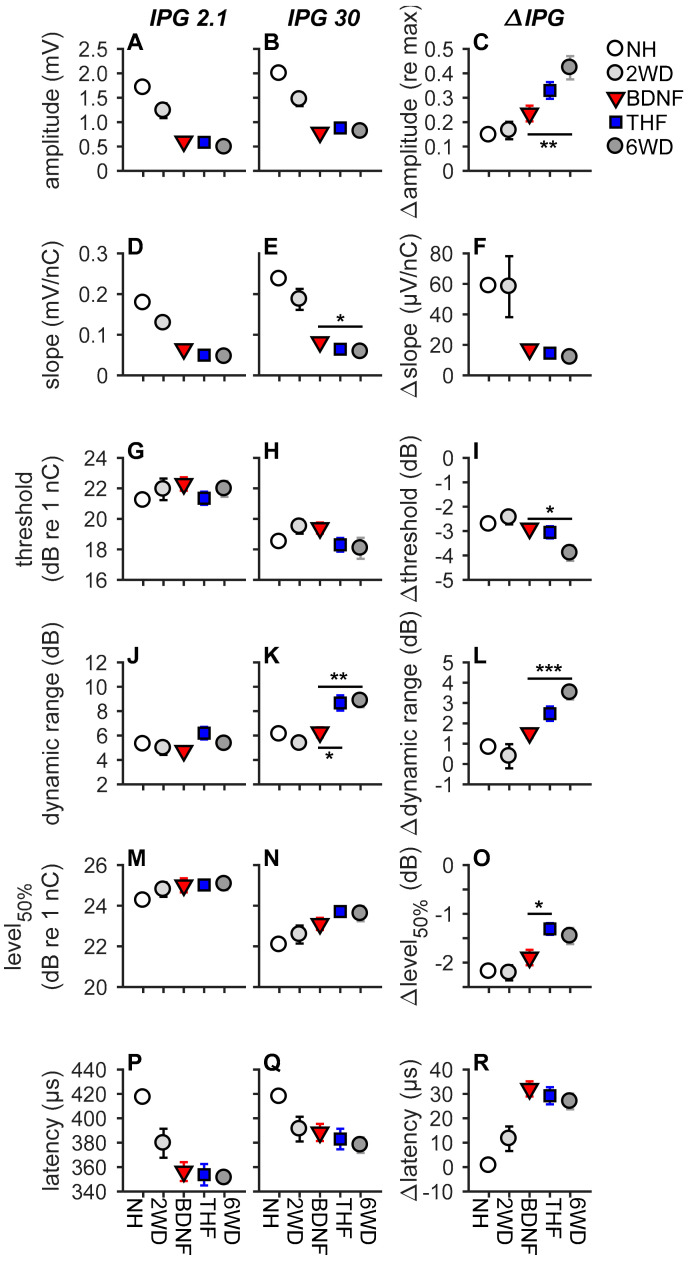
eCAP characteristics for each experimental group. The inter-phase gap (IPG) in the biphasic current pulse was 2.1 µs in the left column and 30 µs in the middle column. The difference between the two (i.e., the IPG effect) is shown in the third column. (**A**–**C**), maximum amplitude; (**D**–**F**), slope; (**G**–**I**), threshold; (**J**–**L**), dynamic range; (**M**–**O**), level_50%_; (**P**–**R**), N_1_ latency. Data of the 2WD animals was used from Ramekers et al. [48]. NH, *n* = 9; 6WD, *n* = 12 (*n* = 13 for latency); BDNF, *n* = 11; THF, *n* = 10 (*n* = 11 for latency). * *p* < 0.05, ** *p* < *0.01,* *** *p* < 0.001. Error bars represent SEM.

**Figure 8 brainsci-10-00787-f008:**
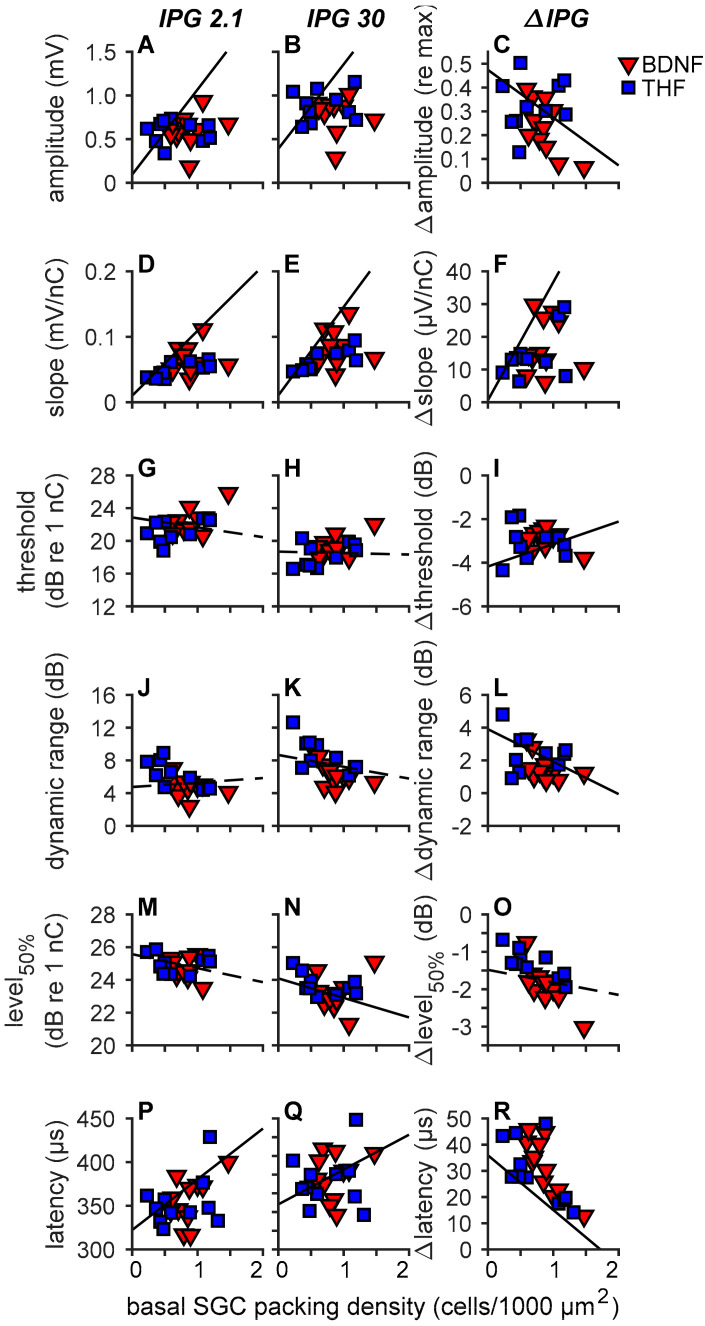
eCAP characteristics of individual animals as a function of basal SGC packing density. The inter-phase gap (IPG) was 2.1 µs in the left column and 30 µs in the middle column. The difference between the two (i.e., the IPG effect) is shown in the third column. Lines represent regression on NH, 2WD (from Ramekers et al. [48]) and untreated 6WD control animals only. Solid lines represent an *R*^2^ with a *p* value of <0.05, dashed lines a *p* value of ≥0.05. (**A**–**C**), maximum amplitude; (**D**–**F**), slope; (**G**–**I**), threshold; (**J**–**L**), dynamic range; (**M**–**O**), level_50%_; (**P**–**R**), N_1_ latency. Data of the 2WD animals was used from Ramekers et al. [48]. BDNF, *n* = 11; THF, *n* = 10 (*n* = 11 for latency). Error bars represent SEM.

**Figure 9 brainsci-10-00787-f009:**
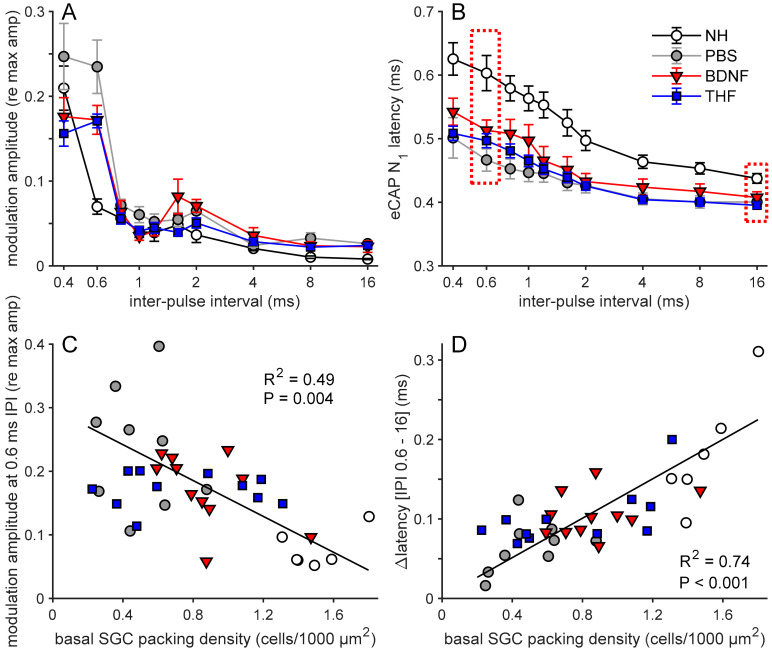
Pulse train response parameters. (**A**) Group means of normalized amplitude modulation determined for the last 6 pulses of a 10–pulse train as a function of IPI. (**B**) Group means of eCAP N_1_ latency averaged over the last 6 pulses of the 10–pulse train as a function of IPI. (**C**) Amplitude modulation for 0.6-ms IPI for individual animals as a function of basal SGC packing density. Regression line and parameters are based on untreated (NH and 6WD) animals only. (**D**) The difference in N_1_ latency between 0.6-ms IPI and 16-ms IPI (dashed boxes in (**B**) as function of basal spiral ganglion cell packing density. Regression line and parameters are based on untreated (NH and 6WD) animals only. NH, *n* = 9 (*n* = 6 in **C**,**D**); PBS, *n* = 9; BDNF, *n* = 11; THF, *n* = 11 animals. IPI, inter-pulse interval. Error bars represent SEM.

**Table 1 brainsci-10-00787-t001:** Results from one-way ANOVAs on single-pulse eCAP characteristics for the BDNF, THF and untreated 6WD groups.

eCAP Measure		IPG 2.1 µs	IPG 30 µs	ΔIPG
Amplitude	*F*	1.4	0.59	5.7
	*p*	0.27	0.56	***0.008 ****
Slope	*F*	3.3	4.1	1.0
	*p*	***0.049 ****	***0.028 ****	0.38
Threshold	*F*	0.9	1.7	4.8
	*p*	0.40	0.19	***0.016 ****
Dynamic range	*F*	2.8	7.5	10.3
	*p*	0.074	***0.002 ****	***<0.001 ****
Level_50%_	*F*	0.013	1.0	3.8
	*p*	0.99	0.37	***0.034 ****
Latency	*F*	0.14	0.50	0.61
	*p*	0.87	0.61	0.55

* Statistically significant *p* value (<0.05).

**Table 2 brainsci-10-00787-t002:** *R*^2^ and *p* values from correlations between single-pulse eCAP characteristics and basal SGC survival in 6WD animals.

eCAP Measure		IPG 2.1 µs	IPG 30 µs	ΔIPG
Amplitude	*R* ^2^	0.56	0.61	0.25
	*p*	***<0.001 ****	***<0.001 ****	***0.014 ****
Slope	*R* ^2^	0.51	0.50	0.29
	*p*	***<0.001 ****	***<0.001 ****	***0.0086 ****
Threshold	*R* ^2^	0.13	0	0.19
	*p*	0.094	0.82	***0.038***
Dynamic range	*R* ^2^	0.05	0.10	0.26
	*p*	0.31	0.14	***0.013 ****
Level_50%_	*R* ^2^	0.17	0.18	0.08
	*p*	0.050	***0.041 ****	0.20
Latency	*R* ^2^	0.50	0.35	0.35
	*p*	***<0.001 ****	***0.0022 ****	***0.0022 ****

* Statistically significant *p* value (<0.05).

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
