# Peer review of "BDNF Outperforms TrkB Agonist 7,8,3′-THF in Preserving the Auditory Nerve in Deafened Guinea Pigs"

_brainsci, 2020, doi:10.3390/brainsci10110787_

Round 1

Reviewer 1 Report

The Manuscript –Brain outperforms TrkB agonist 7,8,3`-THF in preserving the auditory nerve in deafened guinea pigs, reports experimental approaches that tested drugs to prevent auditory nerve damage compromised in cochlear implant users over time.

In guinea pigs naturally occurring neurotrophin brain-derived nerve growth factor BDNF, known to enhance cochlear spiral ganglion SGN survival after damage, in comparison to a small-molecule TrkB agonists THF are analysed. THF is a therapeutic drug predicted to rescue SGN survival through activation of the BDNF receptor TrkB and which has better pharmacokinetic and pharmacodynamics properties in comparison to BDNF.

The study hypothesized that compared to BDNF , the THF compound might lead to better preservation and functionality of cochlear SGCs survival than BDNF itself. As a surprise a contrasting result was found that indicates (i) BDNF to outperform THF in its SGN preservation potential

The Study used an excellent profound study design testing on 45 guinea pigs the hearing function using Click-induced ABR, deafened subgroups through ototoxic aminoglycosides, treated animals single-sided with BDNF or THF 2 weeks post noxious stimuli. 4 weeks later intracochlear electrodes were implanted and hearing tested again through intracochlear electrode recording before animals were sacrified. Finally cochlea were embedded and SGN survival analysed in cross-sections along a standardized midmodiolar plane.

The experimental approaches are written and illustrated in an excellent way that will be well appreciated by a broad audience and scientific community beyond the field.

The topic is of high scientific and clinical interest as a corrects ungoing efforts to use THF as druggable target for improved CI-implant outcome.

Minor revision

Introduction

Line 53: For a journal that it not entirely focused on auditory system specialist, the IPG- effect on eCAP need to be explained in a bit more general context in particular regarding its crucial relevance throughout the study.

Result:

Line 271: The hint to where in the figure what is seen should be at the end of the sentence ? e.g. Figure 5B – should be in line 273..following normal-hearing condition (Figure 5B).

Line 274: Here again…Figure 5C following …without treatment…

Line 285: Here latest the disadvantage of main experimental approaches being solely written in the experimental settings- become evident. The reader is now forced to look again in the experimental settings what is Group B1 and what is Group B2 and what is M1 group etc etc… how many cochlear sections were used to count SGN in which way etc etc.. To overcome this limits- and make the result section more readable for an unfamiliar audience it may be recommendable to (1) either mention in each part of the result once more the most relevant information referring to the specific part in the methods. (2)  Alternatively the result start with a brief chapter that introduce the methodology explained in the detail in the methods parts for Fig 1 to Fig 4.. Meaning e.g. xx animals were deafened, treated with drug and underwent hearing through electrical eCAP recording as described under methods (see Figure 1). eCAP recordings were performed as described under methods and illustrated exemplarily in Figure 2… .

Both is fine- but a more detailed introduction that explains abbreviated words through the entire result part is definitely required to follow the section part with interest.

Line 309: The grouping in B1 , B2, M1 M2, A1 and A2 needs to be explained in the Figure legend of Figure 6.

Line 306: (End of Chapter 3.2 ) As the headings of the result chapter do not give any information about the result- it would extraordinary helpful if at the end of each chapter in the result part a brief summary sentence of the main finding would be included. In the best case this information is used to introduce in the following chapter- of why you now (on the basis of this new finding) were motivated to do the next step..

Line 316: Chapter 3.3 – essential to refer to Figure 2 Line 318. Here it also may be important to briefly explain the difference of IPG and latency… and where – referring to Figure 2 you may see that Latency remains unaffected by varying the IPG (Line 318?)

Line 319: You say the results from the normal-hearing animals and those from the two weeks deaf animals are shown for reference purposes …. Where ? In Figure 7? White or grey dot?? Which panel?? Give more details that the reader can follow..

Line 330 at the end need an information what you refer to? To Table 1 or to Figure 7??

Line 340 : is it the continuance of line 330 or is it the description of Table 1. ?

Line 349- the chapter should end with a brief summarizing sentence of the main result..

Line 401 end of chapter 3.3.3 an Line 429- End of chapter 3.3.4 also a summarizing sentence about the main finding- in words that can be understand by a broad neuroscience community would be helpful.

Discussion

Line 449-to line 458 seems to be a summary introduction of the discussion: Meaning Line 449 to Line 454 could be better switched to the beginning of Chapter 4.1… then you report your finding- and than your surprising finding line 454- fff

Chapter 4.2  Discussion Line 459 ff

It may be most conceivable to integrate a more detailed discussion about what you mean with (line 464) BDNF treatment resulted in more normal-like responsiveness than either THF treatment or the 6WD group ??– for four of these eCAP measures??……Now these four results should be discussed?? (maximal amplitude..latency and level 50% and SGC survival??) What is behind maximal amplitude ?? (what defines the maximal amplitude of ABR wave I? what is currently discussed about it and which auditory fiber subtypes may be involved…?.. what is behind latency… (which auditory fiber..? )  What does it mean that BDNF might improve these paradigms?.Not much is needed here… but the reader may expect- that the effect of BDNF and its difference to THF effect is discussed a bit on what your different experimental readouts you choose may mean for the distinct physiological cochlea function.

Reviewer 2 Report

The study by Vink and colleagues describe the effects of TrkB agonists (its natural ligand BDNF or 7,8,3’THF) on auditory neurons survival and function following acute deafening of guinea pigs with aminoglycosides. This is an important question in the field since cochlear implants performance directly depends on auditory neuron density and function. Therefore, BDNF and other synthetic TrkB agonists might improve cochlear implant function in patients with auditory neuropathy, which are generally poor responders with no possible solutions to recover hearing. The main conclusion of the authors is that when applied from gelatin-sponge BDNF indeed protected auditory neurons while the synthetic trkB agonist 7,8,3’THF was poorly efficient.    

Although the study is overall well conducted with appropriate methodology and relevant preclinical model, I think that a few points need to be clarified.

1- The treatment with neurotrophins was done 1 month prior to assess auditory neuron function and survival. Are there any evidences that such molecules can be active over 1 month from gelatin sponge in vivo? Do the authors know what are BDNF and THF approximate half life in the cochlea? I think this is important to mention existing literature on the topic (if any). It would have been significant added value to the presented functional data if the authors could have titrated the THF or BDNF in the perilymph of guinea pigs at the time of sacrifice.

2- For direct comparison of their effect, critical parameters which are intrinsic to BDNF and THF have to be considered. The problem of solubility is already discussed in the manuscript. I mentioned above the half life of the compounds but their kinetics of release from the sponge is also important. Are there existing prior in vitro data about this? Furthermore, the authors mention line 125 that in the treatment solution, BDNF concentration is 6.67ug/ul while THF concentration is 100 times less. Is the molarity equivalent? This point is crucial for the validity of the conclusion. If concentration is indeed different the conclusion statements might need to be revised.

Minor details:

the procedure looks quite invasive. Has any analgesia been administered to animals? If so, this needs to be mentioned.

Line 173 : for clarity, WD and NH need to be defined earlier in the legend, not at the bottom of the paragraph
